# Young women's reproductive health conversations: Roles of maternal figures and clinical practices

Nicole K. Richards[1,2,3], Elizabeth Crockett[4], Christopher P. Morley[1,2,5], Brooke A. Levandowski[1,6]*

1 Department of Family Medicine, State University of New York Upstate Medical University, Syracuse, New York, United States of America, 2 Department of Public Health and Preventive Medicine, State University of New York Upstate Medical University, Syracuse, New York, United States of America, 3 School of Public Health and Health Systems, University of Waterloo, Waterloo, Ontario, Canada, 4 REACH CNY, Incorporated, Syracuse, New York, United States of America, 5 Department of Psychiatry, Upstate University Hospital, Syracuse, New York, United States of America, 6 Department of Obstetrics and Gynecology, University of Rochester, Rochester, New York, United States of America

* Brooke_Levandowski@URMC.Rochester.edu

**Data Availability Statement:** All relevant data are within the paper and its Supporting Information files.

## Abstract

### Objective

To explore the role of clinical providers and mothers on young women's ability to have confidential, candid reproductive health conversations with their providers.

### Methods

We conducted 14 focus groups with 48 women aged 15–28 years (n = 9), and 32 reproductive healthcare workers (n = 5). Focus groups were audio recorded and transcribed. Data were analyzed using inductive coding and thematic analyses. We examined findings through the lens of paternalism, a theory that illustrates adults' role in children's autonomy and wellbeing.

### Results

Mothers have a substantial impact on young women's health values, knowledge, and empowerment. Young women reported bringing information from their mothers into patient-provider health discussions. Clinical best practices included intermingled components of office policies, state laws, and clinical guidelines, which supported health workers' actions to have confidential conversations. There were variations in how health workers engaged young women in a confidential conversation within the exam room.

### Conclusions

Both young women and health workers benefit from situations in which health workers firmly ask the parent to leave the exam room for a private conversation with the patient. Young women reported this improves their comfort in asking the questions they need to make the best decision for themselves. Clinic leadership needs to ensure that confidentiality

**Funding:** This work was funded, in part, by the Society of Family Planning Research Fund [SFPRF9-CBPR2] and a Research Pilot Project Award from the Department of Obstetrics and Gynecology at the University of Rochester, both of which were received by BAL. The content is solely the responsibility of the authors and does not necessarily represent the official views of the Department of Obstetrics and Gynecology or the University of Rochester. The Society of Family Planning Research Fund provided support in the form of salaries for all authors (NKR, EC, CPM, BAL). Neither funder had any role in the study design, data collection and analysis, decision to publish, or preparation of the manuscript. The specific roles of these authors are articulated in the 'author contributions' section. There was no additional external funding received for this study.

surrounding young women's reproductive health is uniform throughout their practice and integrated into patient flow.

## Introduction

A variety of factors, ranging from clinical best practices to influential parental perspectives, enable an environment for young women to make informed sexual health decisions. Clinical best practice includes incorporating recommendations from professional organizations and legal restrictions. Professional organizations including Society for Adolescent Health and Medicine (SAHM), American Congress of Obstetrics and Gynecology (ACOG), American Academy of Pediatrics (AAP), American Academy of Family Physicians (AAFP), American Medical Association, and many more have advocated for providing a confidential space for young patients to explore topics, usually sexual health-oriented, with their clinician [1–4] which, in 2002, resulted in an amendment to HIIPA that further protected minor privacy in the healthcare setting [5]. SAHM, ACOG, and AAFP recommend providing confidentiality in the form of private conversations as well as billing and medical records to minors to encourage autonomy, healthcare utilization, and sensitive conversations (1–3). SAHM and AAFP recommend providers discuss the importance of privacy with patients and their family, and support communication between minors and their guardians (1,3). ACOG urges providers to be familiar with their state's statutes so office procedures may protect adolescent privacy and encourages provider advocacy for mitigation of state policies that hinder the confidentiality of minors (2). AAP supports adolescent reproductive health confidentiality, specifying that private provider-minor conversations establish rapport and lead to more comprehensive sexual health information and contraceptive care [6].

State laws that limit the ability to consent based on age or legal status create obstacles for providers in ensuring minors' privacy in the healthcare setting [7]. Further, it is not always clear to adolescents what rights they have to confidentiality [8], which differs by state [9]. In New York State, when a minor can consent to treatment, their confidentiality is maintained. However, only minors that have a special status (e.g. pregnant, emancipated) are permitted to make all or most of their own health decisions while other "mature" minors (those deemed to understand benefits and risks of medical care options) can consent to certain services such as reproductive, mental health, addiction, and sexual assault treatments [10]. Confidentiality protocols not only preserve young women's privacy but extend to facilitate young women's comfortability using reproductive healthcare services [11] and divulging sensitive information [12].

Though clinical guidelines support patient confidentiality and rights within a clinical encounter, providers may also consider the role of parents in young women's health decisions. Due to power dynamics and providers' role as medical knowledge gatekeepers, reproductive medicalization is capable of coercing women's health decisions [13,14]. The mothers' third-party advocacy in the reproductive healthcare setting may deter patient exploitation. Research has shown the important role of parental figures in influencing adolescent sexual health behaviors [15,16] and parent-based sexual education successfully leading to increased communication within the parent-child dyad [17]. Not surprisingly, mothers reported the importance of maternal-adolescent conversations about contraception [18] and were identified by young women as a key player in the role of contraceptive method choice, influencing intrauterine contraception uptake [19]. Moreover, evidence has shown that positive maternal relationships were a protective factor against inconsistent condom use [20].

Therefore, while evidence exists of positive maternal influences, challenges remain in the context of familial conversations about sexual activity and birth control. Adolescents reported that their families' attitudes toward sex were more negative than their own as well as their peers' attitudes [21]. In addition, parents report hesitation in discussing sexual health with their children due to their lack of confidence in sex knowledge and concern that discussions encourage sexual behavior [22]. These hesitations can also result in parents making uninformed reproductive health choices for their daughter. For example, young women commonly forego the HPV vaccination due to parental cultural beliefs or mistrust of the medical community [23].

This article is a result of secondary analyses of qualitative data from a study conducted to determine young women and health workers' recommendations to improve contraceptive counseling conversations. Data analyses revealed an inductive theme on how mothers influence their daughters' sexual health decisions. What began to emerge is a picture of paternalism–in the broad parental sense–articulating the complex underlying forces surrounding confidentiality and decision making of young women in the healthcare setting from the perspective of young women and their healthcare teams. Paternalism refers to how policy and norms hinder minors' individual freedom with the intention of preserving wellbeing, and as described by Cohen, "allows adults ownership of children's higher level interests and ultimately segregates children, confining them to the private realm of the family and excluding them from public affairs." [24]. Harms that stem from parental authority upon the minor span a wide variety of illness categories, including mental health [25], vaccines [26], blood transfusion decisions [27], general development of adult autonomy [28], and contraception counseling [29]. In addition to parents' paternalistic influence, healthcare providers also engage in swaying patient decisions through medical paternalism. That is, providers have the power to offer or omit information to a patient with the intention of promoting a certain outcome regardless of the patient's preferences [30]. Thus, a young person's health choices can be manipulated by opposing views of what parents and health professionals think is best for the patient.

Despite recent calls from AAP to amplify the role of the parent in pediatric treatment [31], this is largely a hedge against medical paternalism imposed upon families, as opposed to an argument about the possible harms of paternalism that is realized when parents make decisions for people who have demonstrable autonomy and decision-making capacity, as is often the case with late adolescent patients, and is assumed in the case where the patient has achieved age of majority (typically 18 years of age). Clinical encounters must follow rigorous patient-centered decision-making models to mitigate the power of paternalism and medical paternalism in patient's bodily autonomy [30]. This paper extends the scope of paternalism by including the implicit influence of parents on young women, regardless of adult status. Due to this study's participants expressing particularities of *maternal* influence in the reproductive healthcare setting, we focus specifically on *mothers'* role in the reproductive health decisions of their daughters.

## Methods

This collaborative community based participatory research included a Community Advisory Board (CAB) and hour-long focus groups in a central New York State urban setting. The CAB participated in ten meetings over a three-year period, from designing the research plan to planning actionable steps based on study findings [32]. These meetings identified local contraception counseling dynamics, developed and revised focus group questions, and assisted with knowledge translation. More detail about the CAB's role in this study is available (32).

We conducted seven focus groups with a purposive sample of 48 young women (defined as those aged 15–28 years, to include college-aged women) who will have or who have had contraceptive conversations with health workers. We recruited participants through personal communication from a local sexual health educator (who was also a CAB member), and fliers at a local community agency and community college- both of which also served as focus group locations. Flyers and consent forms contained information about participant inclusion criteria, and researchers verbally confirmed with participants that they will have or have had birth control conversations with a provider. In the first five focus groups, we stratified women by age (15–19,20–24), and race/ethnicity (White non-Hispanic, African American non-Hispanic, and Hispanic [White, African American, other]) (Table 1) to examine differences within a community that faces racial and ethnic disparities in sexual and reproductive health such as teen pregnancy and poor birth outcomes [33]. Women self-identified age, race, and ethnicity. As preliminary data analyses did not find differences in deductive themes by groups, we conducted two final focus groups at a local community college with participants aged 18–28 years, with no racial or ethnicity distinctions. We specifically probed about the influence of maternal figures and factors that made the clinical visit comfortable. We conducted five focus groups with 32 health workers who play a role in contraceptive counseling. Health workers were recruited via CAB connections and practiced in community health and outpatient settings, holding positions that ranged from doulas to intake personnel to Nurse Practitioners. All focus groups contained 2–9 participants. All participants gave verbal informed consent to maintain anonymity; we also obtained verbal informed consent from a parent/legal guardian of those under 18 years old. The consent form contained information about the research (facilitator's experience and study aims), accommodations, and contact information of the research institution. After discussing the consent form, no participants refused or discontinued involvement in this study. This consent process and content was approved by the university's institutional review board.

Focus groups were facilitated by a female epidemiologist (BAL) and documented by a female graduate research assistant (NKR). Both received training in qualitative methods prior to the focus group facilitation. The facilitator verbally provided a background of her previous research initiatives to participants. Focus group questions were revised by the CAB and the resulting focus group guide was approved by the IRB. We asked young women where they obtained information about general health, sexual health, and relationships. We also asked both sets of focus groups about their positive and negative experiences having clinical contraceptive conversations and their recommendations for improving those experiences. We documented field notes and debriefed after each focus group. A hired transcriptionist transcribed all recordings. Transcripts were not reviewed by participants to avoid further participant labor and disruption of the focus groups' observed social process [34]. We coded and

**Table 1. Demographic information on young women focus group participants.**

| Young Women | White (Non-Hispanic) (n) | African American (n) | Hispanic (n) |
|---|---|---|---|
| **Aged 15–19** | -- | 6 | 5 |
| **Aged 20–24** | 3 | 10 | 6 |
| **Aged 18–28**\* | 27 | | |

\*Participants aged 18–28 were recruited after initially stratified focus groups, as initial results did not indicate a different experience between race/ethnicity.

analyzed using inductive and deductive coding and thematic analyses (NVivo v.11). Thematic analysis proceeded in four steps of coding, clustering, subsuming particulars into the general, and confirming. Coding refers to making detailed categories for quotations in direct response to focus group questions or that emerge from participant dialogue. Clustering involves grouping codes of similar themes. Subsuming particulars into the general establishes broad themes while confirming reached consensus on themes between coders [35,36]. Three researchers developed a codebook using an iterative process. NVivo's coding comparison query was used to ensure coding fidelity. Various features of NVivo (examples: word trees, dynamic models, node summary) were used to document the analytical pathway and identify saturation and discrepancies in coding for discussion. The paternalism theory emerged from saturated discussions of mothers within a code entitled "Friends and Family". Thus, the sub-code "Mother" was developed, including all quotations where a participant mentioned the role of a maternal figure.

Focus group findings were presented multiple times to the CAB during the study period; their interpretations influenced study conclusions and future directions. The CAB included those who provide contraception services, youth, public health professionals, and additional community members [32]. CAB members discussed ethical dilemmas of including and excluding maternal figures from reproductive health appointments and expanded on mothers' role in reproductive health outside of the healthcare setting. The insights of diverse community members contributed to the interpretation of the data from the perception of sexual health advocates, mothers, and youth. This study was approved by a Medical University's Institutional Review Board.

## Results

Young women and health workers discussed the complications of confidentiality and how these factors impacted patient-provider reproductive health. Young women considered mothers as a source of information and access to healthcare; however, both young women and health workers shared concerns of incorrect sexual health information stemming from these conversations. Moreover, young women and health workers valued patient privacy in the healthcare setting, but health workers were hesitant in their ability to remove parents from sensitive conversations. Clinical best practices included intermingled components of clinical office policies, state laws, and clinical guidelines, which influenced the ability for patients and providers to have confidential conversations. CAB members reaffirmed these findings and offered potential solutions.

### Maternal figures as an information source

Young women considered maternal figures as a source of advice, information, and resources. They reported that their mothers helped connect them to community and healthcare resources. Acquiring sexual education from community and school settings was dependent on their parents' permission and support. As a 15–19-year-old African American woman stated, "I took all of my classes here [local community center], and I took like one when I was in 7th grade maybe. My mom always made me come here. . .so I used to come to all of them here. Even if they were repetitive, I still came."

Young women discussed their perceived validity of the information they received from maternal conversations. Many had high regard for their maternal figures' health advice, citing life experiences and professions (Licensed Practical Nurse, Registered Nurse, etc.) that made them credible. Though most valued their mothers' health information, some questioned the validity of information that maternal figures provided about sexual education and advice due

to the dynamics of the mother-daughter relationship. One African American 20–24 year old explained, "[My mother's advice] doesn't ever make sense, because I know what she's saying is just for my sake, you know. So it's just like, 'Are you telling the truth or you just want me to hear it?' you know?"

It is important to note that while the majority of young women and health workers reported that having confidential sexual health conversations was desirable to ensure that young women had the information and nonjudgmental space to make sexual health decisions, some examples were given of maternal figures playing supportive roles in sexual health. Some shared stories of observed mother-daughter interactions surrounding health and commented on the comfortability of these situations. Health workers also noted that parents called to provide helpful information and ask questions to support their daughters: "An example [is]. . . Mom calling to say, 'I know she's not taking them regularly as she supposed to.' And there's a concern knowing that she's either definitely or most likely sexually active but not using the birth control that has been provided [accurately]." Also, some young women found comfort in their mother's presence in the exam room. For example, a white 20-24-year-old woman stated, "I'm not good with questions, so that's why I usually have somebody with me at the doctors, like my mom".

However, the majority of health workers recognized that young women confide in and seek advice from them, as people young women trust and respect. Health workers discussed the lack of parental education on the benefits and risks of contraceptive methods, especially new methods and the updated IUD options, or even how to be a reproductive health consumer. Several health workers confirmed young women's concerns of misinformation that they addressed and corrected during clinical visits. One health worker reported, "Some parents don't have that conversation [about sexual education and patient self-efficacy] because they're not expecting their child to have sex at a young age. It happens but they don't want their child to be the ones doing it."

## Need for privacy

Young women valued their providers' promise to ensure confidentiality. They reported trusting their providers with sensitive information, feeling supported by nurses, and more comfortable when health workers assured their protected privacy. As one college student explained, "[The provider] will ask me 'Are you sexually active? This [is] confidential.' I'm like, 'Yes I am sexually active.' Before my mom ever knew I was sexually active. . .that was like a peace of mind that I know my information is not going to get back to my mother because it is confidential, so I had no problem with that.".

Many young women reported being uncomfortable talking to their maternal figures about their own sexual experiences. They feared the consequences of honest dialog such as losing their social freedom. As one 15-19-year-old African American woman said, "Even though I just turned 18, it's like she's [my mother] still like wants to over protect me. And it's just uncomfortable. I'm afraid that she might say I can't like go certain places once I tell her about what's going on in my relationship."

Young women expressed concern with confidentiality issues surrounding insurance and billing information. Patients asked their providers how to overcome these barriers. One college student said, "You can get those confidential services, but you do have to talk with your provider about the way that they bill". Health workers noted that their billing staff took care in supporting patient confidentiality by ensuring that billing information would not go to parents and asking young women if there is another electronic or mailing address they could use for Explanation of Benefits forms.

## Provider responsibility in preserving confidentiality

Clinical best practices include intermingled components of clinical office policies, state laws, and clinical guidelines. These components work either together or against each other to create an environment where confidential patient-provider conversations could occur. While health workers were generally aware of best practice recommendations to allow for a confidential conversation about sexual health [1,37], it became clear from focus groups with both young women and health workers that these recommendations are not uniformly implemented. Both groups reported a variety of scenarios: 1) provider firmly asks the parent to leave for a private conversation; 2) provider asks parent to leave, but allows for negotiation, after which some parents stay; and 3) provider asks the patient if they want the parent to leave in front of the parent, which results in patients either saying "yes" or "no". The last two scenarios were particularly problematic from the standpoint of both young women and health workers. The majority of young women reported that they wanted a private clinician discussion, but negotiations and the fear of stating their desire for a confidential conversation meant that not all young women who wanted a private clinical discussion actually had one. Young women felt pressured to include their mother and thought that asking their mother to leave the providers' office for a private conversation may indicate that they are keeping secrets. Some felt that they were unable to answer questions about sexual health truthfully when their mother was present.

In one focus group of 18–28 year old college students, several women agreed that it should be the providers' responsibility to tell maternal figures to leave during private conversations. One stated, "My mom always gives me the stare, so. . . I know what answer she wants. And it's not like I can go against it." Another agreed, saying "The doctor should be like, 'Alright, parent, please step out of the room,' because if you're like, 'Do you want your parent to leave?'. . .then the parent's like, 'You're hiding something from me.' So then they like dig at you and find out. . ."

Health workers discussed the challenges between their ethical responsibility to ensure the patient has their confidentiality protected, and the sometimes-difficult conversation required to exclude parents from the exam room: "The teen is our patient . . .For parents, it's sometimes hard to come to our office and to relinquish that [decision making] role. . .but when they leave our office, they're still a child and that's their parent, so it's hard to say, 'You don't have to do what your parent wants you to do.''''

Many nurses felt strongly supported throughout their clinic to discuss contraception with every patient in a confidential manner, citing that they recognized when young women do not feel comfortable talking about sexual health with their mothers in the exam room. While in the minority, some health workers were unsure of their office procedures on the implementation of confidentiality laws. As one nurse explained, "I don't know what's ever discussed . . .I mean, I know that there are specific questions that they [MD or NP] ask the patients, but what specifically is discussed when the parent is out of the room, I don't know, because I've never been in the room."

Health workers observed that some maternal figures had strong opinions on which contraceptive method would be the best fit for their child. One health worker stated, "I think one of the problems that I've run into. . .are the very young girls that come in and say, 'My mother says I have to get on Depo' and they really don't want to.'". In response to these situations, health workers reported they focus on empowering young women to make their own individual choices, which they felt resulted in more informed reproductive health decisions. For example, one nurse mentioned ". . .so I tell them, 'Come back without your mom. . .and we can do the appointment, you know, with just you. . ..and we'll find a way for you to conceal whatever [contraceptive method] it is'".

## Community Advisory Board interpretations

Overall, the CAB agreed that sexual education should begin at a young age, involve parental figures, and ensure dissemination of evidence-based information. The CAB discussed a variety of family dynamics and parental characteristics that influence how mothers contribute to the counseling experience of a young woman. They felt that youth and their parents need to be informed about federal and state laws permitting reproductive health autonomy to minors, which, in turn, could support young women's ability to have private contraceptive conversations with their providers. Community members deliberated conflicting focus group data about if providers should request parents to leave the exam room to have a private conversation, or if providers should ask the young women to choose whether they want a private conversation. As was mirrored in the focus group data, CAB members did not achieve consensus on this topic.

The CAB discussed the benefits and shortcomings of community and school-based sexual education. Some adolescents have been excluded from community and school-based sexual education because they did not obtain permission from their guardians or have dropped out of the school system. Considering further research or development of an intervention for young women, the CAB posed the question "How do we assure parents that what we [community health educators] are teaching is accurate and beneficial to their child?" They also suggested that focus groups with guardians of minors should be conducted to understand the parental perceptions on reproductive health and sexual education.

## Discussion

Paternalistic power influences young women's ability to have confidential and effective reproductive health conversations in myriad and complex ways. As suggested in the best practice guidelines (2), young women and health workers both expressed the importance of confidential sexual health conversations, while recognizing the value of parent involvement. Maternal figures were an agent of reproductive knowledge and decisions. This concurs with literature in both national and urban settings which discussed the existing influences of peer and familial relationships on contraceptive uptake and sexual education [38,39].

We showed that young women who preferred their mothers' presence in the reproductive health appointment felt empowered to ask the provider questions. In support of this theme, health workers highlighted the positive impacts supportive parents can have on making informed decisions. A 2010 systematic review of parental influences on adolescent contraceptive decision making discussed the importance of parental communication, connectedness, and approval in delaying sexual activity and promoting healthier sexual behaviors [15]. However, young women and health workers provided more insight on how mothers negatively influence reproductive health conversations than they did on how mothers help facilitation of sensitive information. This finding is supported by literature suggesting that confidentiality protocols improve patient comfortability and honest patient-provider dialogue [11,12]. However, this study's contradicting findings of mothers as a facilitator *or* barrier to young women's informed reproductive health decisions can be explained by paternalism's role in parent-child agreement and concurring autonomy expectations' impact on adolescence adjustment [28]. Since existing clinical guidelines all support protecting adolescent confidentiality, more effort needs to be made to ensure that the confidentiality of young women is prioritized and uniformly implemented [7]. Clinic leadership needs to ensure that all staff, from the nurse and clinic managers to the intake person, know their clinic's stance on supporting young women's rights to confidential sexual health conversations. Various strategies can be employed to integrate this value throughout the clinic, supporting efficient clinic flow.

Within the focus groups, several health workers discussed ways to have a confidential conversation or prescribe contraception to young women that they identify as having an

unsupportive family network. Researchers interpreted these data as an example of the healthcare teams' value for adolescent reproductive health privacy. These findings are similar to a national survey of obstetricians and gynecologists showing 95% of physicians would likely provide oral contraceptives to a young woman without notifying her parents [40]. Further, the CAB discussed possible solutions to avoid tension between young women and their mothers when a provider asks the patient if they would prefer a confidential conversation. The suggestion already mentioned by both young women and health workers to have time dedicated to talking to young women in private was reiterated by the CAB. By taking this approach, young women have more autonomy on choosing how they participate in reproductive health conversations.

As with all qualitative studies, these focus groups represent the thoughts and feelings of participants in one community. However, these findings identify barriers and facilitators to young women's confidential reproductive health care and inform protocol/policy on the implementation of clinical best practices supporting adolescent confidentiality. We did not hold any focus groups with physicians, which often act as prescribers and counselors to young women. The perceptions and experience of midlevel and other health workers who were interviewed offer a valuable contribution to the literature, as they may have more opportunity to build rapport with patients due to physician time constraints. Further, this research used paternalism as an emergent theory. Therefore, inductive investigation may have missed detailed information about the maternal role in patient-provider contraception conversations. Specifically, the study design is limited by secondary analysis because it did not include paternalism driven focus group guides or focus groups with maternal figures of young women.

In addition to exploring how maternal figures and clinical best practice influences young women's navigation of reproductive healthcare, we showed that these factors contribute to young women's autonomy in making reproductive health decisions. Clinical recommendations about adolescent reproductive healthcare coincide with this study's findings, highlighting the importance of the provider ensuring confidential space for having confidential sexual health conversations with young women. Though New York State law protects the reproductive health confidentiality of most adolescents [10], not all young women are given an opportunity for a private patient-provider conversation. To increase young women's confidentiality in the healthcare setting, more research is necessary to reach other stakeholder input, including gathering the perspective of reproductive health physicians and maternal figures. Further research may be designed and implemented through theories that focus on paternalistic influences on clinical health decisions.

## Conclusion

Both young women and providers benefit from situations in which providers firmly ask the parent to leave the exam room for a private conversation with the patient. Young women reported this improves their comfort in asking the questions they want, to obtain the information they need to make the best decision for themselves. Clinic leadership needs to ensure that confidentiality surrounding young women's reproductive health is uniform throughout their practice and integrated into patient flow. Providers should implement a variation of a shared decision making model to mitigate the possibility of paternalistically imposing on young women's health decisions [30]. These strategies can improve the efficacy of reproductive health counseling and autonomy among young women.

## Supporting information

**S1 File. COREQ checklist.**
(PDF)

## Acknowledgments

We appreciate the time and effort of focus group participants and the Community Advisory Board. We would also like to thank Desirree Pizarro, MPH for her contribution to data collection and analysis.

## Author Contributions

**Conceptualization:** Elizabeth Crockett, Brooke A. Levandowski.

**Data curation:** Nicole K. Richards, Brooke A. Levandowski.

**Formal analysis:** Nicole K. Richards, Brooke A. Levandowski.

**Funding acquisition:** Elizabeth Crockett, Christopher P. Morley, Brooke A. Levandowski.

**Investigation:** Brooke A. Levandowski.

**Methodology:** Brooke A. Levandowski.

**Project administration:** Christopher P. Morley, Brooke A. Levandowski.

**Resources:** Christopher P. Morley, Brooke A. Levandowski.

**Software:** Nicole K. Richards, Christopher P. Morley.

**Supervision:** Brooke A. Levandowski.

**Validation:** Brooke A. Levandowski.

**Visualization:** Brooke A. Levandowski.

**Writing – original draft:** Nicole K. Richards.

**Writing – review & editing:** Elizabeth Crockett, Christopher P. Morley, Brooke A. Levandowski.

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
