## [Decision Letter · Decision Letter 0]

12 Sep 2019

PONE-D-19-19922

Young women’s reproductive health conversations: Roles of maternal figures and clinical practices

PLOS ONE

Dear Dr Levandowski,

Thank you for submitting your manuscript to PLOS ONE. After careful consideration, we feel that it has merit but does not fully meet PLOS ONE’s publication criteria as it currently stands. Therefore, we invite you to submit a revised version of the manuscript that addresses the points raised during the review process.

More specifically, the reviewers and I are in agreement that as currently written, it is unclear that the manuscript meets the following PLOS ONE publication criteria:

Experiments, statistics, and other analyses are performed to a high technical standard and are described in sufficient detail.Conclusions are presented in an appropriate fashion and are supported by the data.

See specific comments from the reviewers and I regarding this and other matters below and attached. 

We would appreciate receiving your revised manuscript by Oct 27 2019 11:59PM. To enhance the reproducibility of your results, we recommend that if applicable you deposit your laboratory protocols in protocols.io, where a protocol can be assigned its own identifier (DOI) such that it can be cited independently in the future. For instructions see: http://journals.plos.org/plosone/s/submission-guidelines#loc-laboratory-protocols

We look forward to receiving your revised manuscript.

Kind regards,

Whitney S. Rice, DrPH

Academic Editor

PLOS ONE

Journal Requirements:

2. Please amend your current ethics statement to address the following concerns: Please explain why was written consent was not obtained, how you recorded/documented participant consent, and if the ethics committees/IRBs approved this consent procedure.

3. Please include a copy of the interview guide used in the study, in both the original language and English, as Supporting Information, or include a citation if it has been published previously.

 [This work was funded, in part, by the Society of Family Planning Research Fund [SFPRF9-CBPR2] and a Research Pilot Project Award from the Department of Obstetrics and Gynecology at the University of Rochester, both awarded to BAL.  The content is solely the responsibility of the authors and does not necessarily represent the official views of the Department of Obstetrics and Gynecology or the University of Rochester.]. 

[No authors have competing interests].   

We note that one or more of the authors are employed by a commercial company: 'REACH CNY, Inc'.

Additional Editor Comments (if provided):

l agree with the reviewer 2 that the introduction and discussion in particular seem to miss important contextual literature on mothers' and providers' involvement in the healthcare interactions of young women. The paper largely does not engage with (or even acknowledge) the tension between the benefits of confidentiality for young women on one end, the potential for medical providers to limit patient autonomy (and enact paternalism) on the other end, and then the potential for mothers or other family members to facilitate patient autonomy and act in the best interest of young women on the other end. Consider drawing upon the extant literature regarding family engagement in healthcare and patient-provider relationship dynamics (including paternalism) in this context. How did the authors analyze their findings "through the lens of paternalism"? This doesn't come through clearly in the methods or results.Please ensure that your description of qualitative methods meet the requirements of the COREQ checklist (attached) or other qualitative reporting checklist appropriate for your professional discipline.

Reviewers' comments:

Reviewer's Responses to Questions

**Comments to the Author**

1. Is the manuscript technically sound, and do the data support the conclusions?

Reviewer #1: Yes

Reviewer #2: Partly

2. Has the statistical analysis been performed appropriately and rigorously? 

Reviewer #1: N/A

Reviewer #2: Yes

3. Have the authors made all data underlying the findings in their manuscript fully available?

Reviewer #1: Yes

Reviewer #2: Yes

4. Is the manuscript presented in an intelligible fashion and written in standard English?

Reviewer #1: Yes

Reviewer #2: Yes

5. Review Comments to the Author

Reviewer #1: This manuscript was well written, clear, and concise. The demographic table (Table 1) was confusing in that the age groups overlapped and using "n=" in each box was redundant. A review of this table and further explanation as to why the age groups overlap would be helpful. Race was not broken down for the 18-28 age group, which was confusing. An explanation as to why this age category was not broken down by race would be helpful.

Reviewer #2: I have uploaded my review as an attachment. For some reason, when I pasted in my comments, it stated that I did not meet the minimum character count.

6. PLOS authors have the option to publish the peer review history of their article (what does this mean?). If published, this will include your full peer review and any attached files.

Reviewer #1: No

Reviewer #2: No

---

## [Author Response · Author response to Decision Letter 0]

29 Oct 2019

Greetings Dr. Rice,

We are pleased to submit our revised manuscript entitled “Young women’s reproductive health conversations: Roles of maternal figures and clinical practices” to PLOS One for your consideration. We have responded to the reviewer comments below, and uploaded a revised submission with track changes marked.

Journal Requirements:

AUTHOR RESPONSE:

We appreciate these guidelines and have made appropriate changes to the manuscript’s headings, citations, and acknowledgement section.

2. Please amend your current ethics statement to address the following concerns: Please explain why was written consent was not obtained, how you recorded/documented participant consent, and if the ethics committees/IRBs approved this consent procedure.

 AUTHOR RESPONSE:

 We have provided further information on the consent process and disclosed the IRB’s approval of this method of consent.

3. Please include a copy of the interview guide used in the study, in both the original language and English, as Supporting Information, or include a citation if it has been published previously.

AUTHOR RESPONSE:

We appreciate your commitment to transparency. Our interview guides has been submitted with another manuscript with this data, offering the main findings of the study, which is currently under review. We would prefer to keep the interview guide published with the main study findings versus a secondary data analyses, and appreciate the timing is not desirable. We appreciate your patience and understanding. 

4. Please provide an amended statement that declares *all* the funding or sources of support (whether external or internal to your organization) received during this study, as detailed online in our guide for authors at http://journals.plos.org/plosone/s/submit-now. Please also include the statement “There was no additional external funding received for this study.” in your updated Funding Statement.* Please include your amended Funding Statement within your cover letter. We will change the online submission form on your behalf.

 AUTHOR RESPONSE: Please find our amended Funding Statement below: 

This work was funded, in part, by the Society of Family Planning Research Fund [SFPRF9-CBPR2] and a Research Pilot Project Award from the Department of Obstetrics and Gynecology at the University of Rochester, both of which were received by BAL. The content is solely the responsibility of the authors and does not necessarily represent the official views of the Department of Obstetrics and Gynecology or the University of Rochester.

The Society of Family Planning Research Fund provided support in the form of salaries for all authors (NKR, EC, CPM, BAL). Neither funder had any role in the study design, data collection and analysis, decision to publish, or preparation of the manuscript. The specific roles of these authors are articulated in the ‘author contributions’ section. There was no additional external funding received for this study.

Please also provide an updated Competing Interests Statement declaring this commercial affiliation along with any other relevant declarations relating to employment, consultancy, patents, products in development, or marketed products, etc. 

AUTHOR RESPONSE: Please see our amended Funding Statement above. No authors had Competing Interests.

Editor.

1. The introduction and discussion in particular seem to miss important contextual literature on mothers' and providers' involvement in the healthcare interactions of young women. The paper largely does not engage with (or even acknowledge) the tension between the benefits of confidentiality for young women on one end, the potential for medical providers to limit patient autonomy (and enact paternalism) on the other end, and then the potential for mothers or other family members to facilitate patient autonomy and act in the best interest of young women on the other end. Consider drawing upon the extant literature regarding family engagement in healthcare and patient-provider relationship dynamics (including paternalism) in this context. 

AUTHOR RESPONSE:

Thank you for this comment. Based on these suggestions, we have added information on A. The role of confidentiality protocols in young women’s health utilization. B. The potential for providers to act as gatekeepers to patients’ health information and decision options. C. Using mothers as advocates for autonomous health choices. D. Connected the influence of both parents and providers to broad paternalism and medical paternalism.

2. How did the authors analyze their findings "through the lens of paternalism"? This doesn't come through clearly in the methods or results. 

AUTHOR RESPONSE:

We appreciate this question. To give more transparency in our analysis through the lens of paternalism, we have elaborated on the steps of our methods by including how paternalism theory emerged from our codes.

3. Please ensure that your description of qualitative methods meet the requirements of the COREQ checklist (attached) or other qualitative reporting checklist appropriate for your professional discipline.

AUTHOR RESPONSE:

We appreciate you providing this checklist, which was attached as supplemental information in the original submission. We have reviewed this list and added more information addressing the rigor qualifications by explaining more about the consent process. We also provide an explanation for the decision not to member-check.

Reviewer 1:

1. The demographic table (Table 1) was confusing in that the age groups overlapped and using "n=" in each box was redundant. A review of this table and further explanation as to why the age groups overlap would be helpful. Race was not broken down for the 18-28 age group, which was confusing. An explanation as to why this age category was not broken down by race would be helpful.

 AUTHOR RESPONSE: 

 We appreciate these observations. Based on these comments, we agree that table 1 was confusing. Therefore, we have removed the redundant “n=” and added a note to the table explaining age and race inconsistencies. We also provide a further explanation within the methodology text. 

Reviewer 2:

1. Is it possible to put the total number of focus group participants in the abstract? Should the concept of paternalism be added to the abstract since it’s a major part of this manuscript?

 AUTHOR RESPONSE:

 Thank you for these abstract recommendations. We revised the abstract to include the number of participants and elaborated on the role of paternalism.

2. Lines 80-87: Why isn’t there a mention about the specific role of father’s? If the

 focus of the manuscript is mothers over fathers, it should be explained why.

 AUTHOR RESPONSE:

 We appreciate this point and have explained why this manuscript focuses on the mother’s role.

3. Line 106: Is parental authority always an imposition? Lines 109-110: “We focus specifically on mothers’ imposition in the reproductive health decisions of their daughters. “ Is this statement too harsh? Are you exploring whether this happens or how it happens? Or are you saying it happens all the time?

AUTHOR RESPONSE:

We appreciate these insightful questions and comments. We have adjusted our wording around this statement. In addition, we have added information throughout the introduction and discussion on both the positive and negative influences of paternalism.

4. Lines 93-110: Could you explain more about the theme of paternalism that you found in your primary analysis of this data? Is paternalism always negative? Could you define medical paternalism? What is the difference between parental paternalism and medical paternalism? How can an adolescent be protected from both parental paternalism and medical paternalism to ensure they are making their own autonomous decision? Or is it even possible? This could possibly be included in the discussion section. Or is medical paternalism important to this manuscript since it was not mentioned in the discussion/conclusion sections? How does amplifying the role of a parent, help against medical paternalism? Could you explain more about paternalism and reproductive health decision-making? Is there a difference between parental paternalism versus medical paternalism in reproductive decision-making? It might be helpful to include how paternalism can hinder (or may help?) the ability of young women to have confidential conversations with their provider.

AUTHOR RESPONSE:

We appreciate these questions and have restructured our introduction, methodology, discussion, and conclusion to give a better picture of how we used paternalism in our analysis as well as paternalism’s negative and positive influences in the decision-making process. We added more information on medical paternalism’s definition and role in decision making. We also included suggestions to mitigate negative influence and how to appropriately include parents.

5. Line 112: Could you explain more about the role of the CAB for your study beyond what is stated in Line 151-154? Why was there interpretation of the data important (Lines 259-274)?

AUTHOR RESPONSE:

Thank you for this question. We have further explained the role of the CAB and added an example of their interpretations suggesting actionable steps to address paternalism in the healthcare setting.

6. Lines 121-122: “Hispanic women could choose a focus group with women of similar race or ethnicity.” Why was this decided?  Why couldn’t other race of women do the same?

AUTHOR RESPONSE:

We appreciate this point. Upon discussion, we agreed to eliminate this sentence as it was redundant since we mention that any participant was able to self-identify their race or ethnicity.

7. Table 1: Could you add the racial breakdown of the final two focus groups (Aged 18-28) and the average age of these focus groups?

AUTHOR RESPONSE:

Thank you for this question. Our qualitative research was iterative in nature. We first conducted focus groups with young women which were divided by age and race/ethnicity. We analyzed this data and found no variation of experiences or perceptions between age or race/ethnicity within the original focus groups. However, we were missing the perspective of college aged young women, prompting the final two focus groups of young women aged 18-28. For these focus groups, we did not collect these participant demographics. 

8. From the results sections it appears that some of the young women did not mind or wanted their mothers to be in the room in their discussions with their providers. In lines 299-201 and lines 319-320, you discussed that health care providers would like to provide oral contraceptives to young women without their parent’s consent and you stated that young women and providers benefit from situations where the provider asks the parent to leave the room. However how should it be navigated when the young woman wants her mother involved? How do we ensure a young woman makes an autonomous decision about her reproductive health whether the mother is in the clinic room or not?

AUTHOR RESPONSE:

We appreciate these questions on how to address this complex issue. We have added the CAB’s suggestions to navigate including and excluding parents in sensitive patient-provider conversations. We have also suggested using protocols that facilitate patient autonomy.

9. Additionally shouldn’t we still have concerns about medical paternalism? For example, if a young woman comes into the clinic and wants birth control pills, but through private counseling with the provider she is convinced to get an IUD, are we sure she made an autonomous decision?

AUTHOR RESPONSE: 

We appreciate this thoughtful question. Upon discussion, we decided it was important to touch more on the effects of medical paternalism throughout the introduction and discussion.

We hope these responses and the subsequent changes to our manuscript fully address the comments. We thank you for this review of our work and look forward to a favorable response.

Sincerely,

Brooke A. Levandowski, PhD, MPA

Assistant Professor

Department of Obstetrics and Gynecology

Clinical and Translational Science Institute

University of Rochester Medical Center

Brooke_levandowski@URMC.rochester.edu

585-275-3727

---

## [Decision Letter · Decision Letter 1]

31 Dec 2019

PONE-D-19-19922R1

Young women’s reproductive health conversations: Roles of maternal figures and clinical practices

PLOS ONE

Dear Dr Levandowski,

Thank you for submitting your manuscript to PLOS ONE, and for your patience in awaiting an outcome. Before we can accept the paper for publication, we would like you to re-submit a final revision which addresses additional editorial comments (listed below), and PLOS ONE’s publication criteria.

Methods• Page 5, Line 127: I agree with Reviewer 2 that extent to which this research is participatory is unclear. Please expand upon the statements around "This collaborative community based participatory research" and the inclusion of a "Community Advisory Board (CAB)". Did the CAB contribute to study design or to the development of study materials? did the CAB directly refer the connections that facilitated recruitment?• The methods section lacks detail about the focus group process. Was discussion facilitated by a focus group guide? Were questions about adolescent women's relationships and their influence on sexual health decisions asked? What questions were asked?• Page 5, Line 128 - Page 6, Line 130: How did the authors assess that women had or would have contraceptive conversations with a provider?• Page 7, Line 170: Please expand upon what is meant by "minute, detailed ratings"Discussion• Should the lack of mothers' perspectives in this study be included as another limitation? How might the study have been enriched by these perspectives?Additionally, how comprehensively could this study have explored paternalism and adolescent sexual health decision-making considering that it is a secondary data analysis?  We would appreciate receiving your revised manuscript by Feb 14 2020 11:59PM. To enhance the reproducibility of your results, we recommend that if applicable you deposit your laboratory protocols in protocols.io, where a protocol can be assigned its own identifier (DOI) such that it can be cited independently in the future. For instructions see: http://journals.plos.org/plosone/s/submission-guidelines#loc-laboratory-protocols

Please include the following items when submitting your revised manuscript:A rebuttal letter that responds to each point raised by the academic editor and reviewer(s). This letter should be uploaded as separate file and labeled 'Response to Reviewers'.A marked-up copy of your manuscript that highlights changes made to the original version. This file should be uploaded as separate file and labeled 'Revised Manuscript with Track Changes'.An unmarked version of your revised paper without tracked changes. This file should be uploaded as separate file and labeled 'Manuscript'.

We look forward to receiving your revised manuscript.

Kind regards,

Whitney S. Rice, DrPH

Academic Editor

PLOS ONE

Reviewers' comments:

Reviewer's Responses to Questions

**Comments to the Author**

1. If the authors have adequately addressed your comments raised in a previous round of review and you feel that this manuscript is now acceptable for publication, you may indicate that here to bypass the “Comments to the Author” section, enter your conflict of interest statement in the “Confidential to Editor” section, and submit your "Accept" recommendation.

Reviewer #1: All comments have been addressed

2. Is the manuscript technically sound, and do the data support the conclusions?

Reviewer #1: Yes

3. Has the statistical analysis been performed appropriately and rigorously? 

Reviewer #1: N/A

4. Have the authors made all data underlying the findings in their manuscript fully available?

Reviewer #1: No

5. Is the manuscript presented in an intelligible fashion and written in standard English?

Reviewer #1: Yes

6. Review Comments to the Author

Reviewer #1: (No Response)

7. PLOS authors have the option to publish the peer review history of their article (what does this mean?). If published, this will include your full peer review and any attached files.

Reviewer #1: No

---

## [Author Response · Author response to Decision Letter 1]

6 Jan 2020

Methods:

1. Page 5, Line 127: I agree with Reviewer 2 that extent to which this research is participatory is unclear. Please expand upon the statements around "This collaborative community based participatory research" and the inclusion of a "Community Advisory Board (CAB)". Did the CAB contribute to study design or to the development of study materials? did the CAB directly refer the connections that facilitated recruitment?

AUTHOR RESPONSE:

We appreciate this question. We have expanded on the role of the CAB (lines 128 – 132) and included a link to our publication about this project’s CAB for more details on their contributions. More information about the CAB’s contributions to materials can be found in lines 167-169 and participant recruitment in lines 135-136, 147-148.

2. The methods section lacks detail about the focus group process. Was discussion facilitated by a focus group guide? Were questions about adolescent women's relationships and their influence on sexual health decisions asked? What questions were asked?

AUTHOR RESPONSE:

Thank you for these recommendations to improve clarity. We have added the term “focus group guide” to our statement about question development in lines 167 – 168. As mentioned in the last revision, we have submitted another manuscript with the main study findings with the study guides included. Therefore, we have not included them here.

3. Page 5, Line 128 - Page 6, Line 130: How did the authors assess that women had or would have contraceptive conversations with a provider?

AUTHOR RESPONSE:

We appreciate this question about inclusion process. We have included more detail on how we received self-reported future and/or past patient-provider contraceptive conversations in lines 137 – 139.

4. Page 7, Line 170: Please expand upon what is meant by "minute, detailed ratings"

AUTHOR RESPONSE:

We agree that this description does not adequately explain this step of thematic analysis. We have changed this sentence to better communicate what we mean by coding (lines 176 – 177).

Discussion

5a. Should the lack of mothers' perspectives in this study be included as another limitation? How might the study have been enriched by these perspectives?

5b. Additionally, how comprehensively could this study have explored paternalism and adolescent sexual health decision-making considering that it is a secondary data analysis?

AUTHOR RESPONSE:

Thank you for these questions. We agree that this research design is limited by paternalism as an emergent theory. We have included lack of paternalism driven questions and mothers’ perspectives as another limitation in lines 351 -354. Please also see lines 361 – 364 for our recommended next steps.

---

## [Editor Report · Decision Letter 2]

9 Jan 2020

Young women’s reproductive health conversations: Roles of maternal figures and clinical practices

PONE-D-19-19922R2

Dear Dr. Levandowski,

We are pleased to inform you that your manuscript has been judged scientifically suitable for publication and will be formally accepted for publication once it complies with all outstanding technical requirements.

With kind regards,

Whitney S. Rice, DrPH

Academic Editor

PLOS ONE
---

## [Editor Report · Acceptance letter]

13 Jan 2020

PONE-D-19-19922R2 

Young women’s reproductive health conversations: Roles of maternal figures and clinical practices 

Dear Dr. Levandowski:

I am pleased to inform you that your manuscript has been deemed suitable for publication in PLOS ONE. Congratulations! Your manuscript is now with our production department. 

With kind regards,

on behalf of

Dr. Whitney S. Rice 

Academic Editor

PLOS ONE